# TreeCUA: Efficiently Scaling GUI Automation with Tree-Structured Verifiable Evolution

**Deyang Jiang** [* 1]  **Jing Huang** [* 1]  **Xuanle Zhao** [* 1]  **Lei Chen** [1]  **Liming Zheng** [1]  **Fanfan Liu** [1]  **Haibo Qiu** [1]
**Peng Shi** [1]  **Zhixiong Zeng** [✉ 1]

## Abstract

Effectively scaling GUI automation is essential for computer-use agents (CUAs); however, existing work primarily focuses on scaling GUI grounding rather than the more crucial GUI planning, which requires more sophisticated data collection. In reality, the exploration process of a CUA across apps/desktops/web pages typically follows a tree structure, with earlier functional entry points often being explored more frequently. Thus, organizing large-scale trajectories into tree structures can reduce data cost and streamline the data scaling of GUI planning. In this work, we propose TreeCUA to efficiently scale GUI automation with tree-structured verifiable evolution. We propose a multi-agent collaborative framework to explore the environment, verify actions, summarize trajectories, and evaluate quality to generate high-quality and scalable GUI trajectories. To improve efficiency, we devise a novel tree-based topology to store and replay duplicate exploration nodes, and design an adaptive exploration algorithm to balance the depth (*i.e.*, trajectory difficulty) and breadth (*i.e.*, trajectory diversity). Moreover, we develop world knowledge guidance and global memory backtracking to avoid low-quality generation. Finally, we naturally extend and propose the TreeCUA-DPO method from abundant tree node information, improving GUI planning capability by referring to the branch information of adjacent trajectories. Experimental results show that TreeCUA and TreeCUA-DPO offer significant improvements, and out-of-domain (OOD) studies further demonstrate strong generalization. All trajectory node information and code will be available at https://github.com/UITron-hub/TreeCUA.

---

[*]Equal contribution  [1]Meituan, Beijing, China. Correspondence to: Zhixiong Zeng <zengzhixiong@meituan.com>.

*Proceedings of the 43rd International Conference on Machine Learning*, Seoul, South Korea. PMLR 306, 2026. Copyright 2026 by the author(s).

## 1. Introduction

Computer-Use Agents (CUAs) represent a paradigm shift towards general-purpose digital agents, capable of perceiving and interacting with Graphical User Interfaces (GUIs) in a human-like manner (OpenAI, 2025). Leveraging the emergent capabilities of large Vision-Language Models (VLMs) (Bai et al., 2025; Comanici et al., 2025), these agents process high-resolution visual inputs to execute intricate instruction-following tasks. The field has been significantly shaped by proprietary frontier models such as Gemini (Comanici et al., 2025) and Claude (Anthropic, 2025), which have demonstrated exceptional proficiency in complex agentic workflows. Concurrently, open-source initiatives are accelerating efforts to scale CUA models and data pipelines to bridge the gap with proprietary systems. Representative models (Qin et al., 2025; Ye et al., 2025; Zeng et al., 2025; Sun et al., 2024) have successfully scaled capabilities across desktop, mobile, and web platforms, underscoring the immense potential for generalized digital intelligence.

However, distinct from the rapid scaling of model parameters, scaling high-quality training data incurs significantly higher costs and complexity. Although recent works (Cheng et al., 2024a; Xie et al., 2025; Chai et al., 2024; Huang et al., 2025) have successfully scaled GUI grounding datasets, these efforts focus primarily on static element recognition. In contrast, efficiently scaling long-horizon planning trajectories necessitates temporal reasoning and dynamic interaction, remaining a largely unexplored challenge. To address this challenge, recent CUA methods like OpenCUA (Wang et al., 2025) and ScaleCUA (Liu et al., 2025) collect human computer-use demonstrations or human expert annotations to construct GUI trajectories. Unfortunately, existing methods often rely on necessary human annotations, making it difficult to scale effectively to large-scale trajectory synthesis. As a result, open-source trajectories currently available in this field remain very scarce.

Efficiently scaling GUI trajectory requires addressing two key challenges: step redundancy and trajectory diversity. Step redundancy intensifies with increasing trajectory scale, as different app/desktop/webpage trajectories inevitably repeat the exploration of early functional entry points. Trajec-

tory diversity depends primarily on the distillation model in automated trajectory synthesis, as the model's inherent bias favors high-frequency exploration behaviors, potentially leading to insufficiently diverse and long-tail trajectories.

In this paper, we devise an efficient way to scale GUI automation by eliminating step redundancy through organizing large-scale trajectories into a tree structure and improving trajectory diversity through world knowledge guidance. Thus we propose TreeCUA, a novel data synthesis framework of tree-structured verifiable evolution, and design a training recipe of two-stage SFT protocol for establishing foundational and cognitive capabilities. To this end, we develop an automated multi-agent collaborative framework to collect tree-structured verifiable synthesis trajectories, including an exploration agent, a verification agent, a summary agent and an evaluation agent to explore the environment, verify actions, summarize trajectories, and evaluate quality. To achieve efficiency and scalability, we devise a novel tree-based topology to store and reuse intermediate exploration nodes. Moreover, we design an adaptive exploration algorithm to balance trajectory depth (difficulty) and breadth (diversity) within the tree exploration, augmented by world knowledge guidance and global memory backtracking to strictly filter low-quality generations.

In particular, leveraging the rich tree-structured data, we naturally extended TreeCUA to TreeCUA-DPO through reinforcement learning.

By utilizing the branching information from adjacent trajectories as distinct preference pairs, this method significantly enhances the model planning capabilities. Experimental results on OSWorld (Xie et al., 2024) and out-of-distribution (OOD) tasks demonstrate that both TreeCUA and TreeCUA-DPO significantly improve task success rates in online environments, while also exhibiting strong generalization capabilities. In conclusion, our contributions are listed as follows

- We introduce tree-structured verifiable evolution for efficiently scaling CUA trajectory synthesis, which develops a multi-agent framework to explore the environment, verify actions, summarize trajectories, and evaluate quality.

- Based on tree-structured trajectories, we propose TreeCUA and TreeCUA-DPO for GUI automation, which devises a two-stage SFT protocol to establish foundational capability and DPO training for referring to the branch information of adjacent trajectories.

- Extensive experimental results show that our methods achieve high performance on in-domain OSWorld and OOD tasks, significantly outperforming existing open-source trajectories and also demonstrating remarkable

generalization.

## 2. Related Works

### 2.1. Computer-Use Agents (CUAs)

The evolution of Computer-Use Agents (CUAs) has progressed through three methodological paradigms, shifting from dependence on metadata to holistic visual perception (Cheng et al., 2024a; Sun et al., 2024). Initially, research primarily relied on structured metadata, such as DOM trees (Deng et al., 2023; Pan et al., 2024), to generate symbolic commands. While effective in structured environments, these text-based models often prove brittle to dynamic web elements or applications that lack accessibility tags. Subsequently, to incorporate visual perception, a modular "Planner-Grounder" framework emerged (Cheng et al., 2024a; Wu et al., 2024a). These systems leverage strong VLMs for high-level planning alongside specialized modules for localization (Yang et al., 2025a; Huang et al., 2025). However, such multi-stage pipelines frequently suffer from high computational latency and error propagation. Most recently, the field has moved towards native end-to-end agents that integrate planning and grounding into a unified model (Sun et al., 2024; Wang et al., 2025; Liu et al., 2025). By directly translating screenshots into executable actions, representative models like Aguvis (Xu et al., 2024), UItron(Zeng et al., 2025), UI-Tars (Qin et al., 2025) and OpenCUA (Wang et al., 2025), achieve superior perception-action alignment.

### 2.2. CUA Data Synthesis

Achieving GUI automation requires strong GUI grounding and planning capabilities. These capabilities allow the GUI agent to pinpoint precise visual elements from screenshots and execute planned actions to complete the user task. Previous research has primarily concentrated on the data collection of GUI grounding, where typical works include SeeClick (Cheng et al., 2024b), AMEX (Chai et al., 2024), OS-Atlas (Wu et al., 2024b), and JEDI (Xie et al., 2025). Recently, GroundCUA (Feizi et al., 2025) collected a large-scale desktop grounding dataset built from expert human demonstrations, and ScaleCUA (Liu et al., 2025) released a large-scale grounding dataset spanning 6 operating systems and 3 task domains, annotated by automated agents and human experts. However, these works are confined to static element recognition, overlooking the sequential reasoning essential for multi-step planning. To support multi-step task planning, researchers have developed various strategies for trajectory synthesis (Rawles et al., 2023; Xu et al., 2025; Sun et al., 2024; Pahuja et al., 2025), utilizing human demonstrations or screen recordings to generate planning trajectories in domains like android control and web agent. Recently, OpenCUA (Wang et al., 2025) proposes an anno-

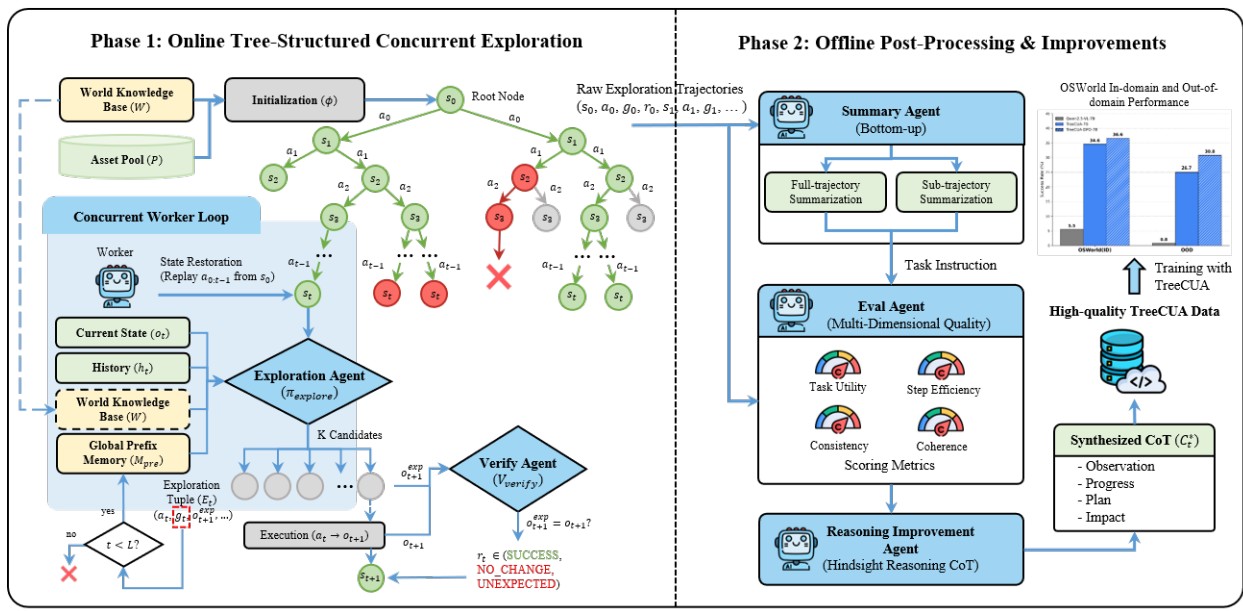

**Figure 1.** Overview of the tree-structured verifiable evolution for scalable GUI trajectory synthesis. Our strategy could be divided into online concurrent exploration and offline post-processing and improvement phases.

tation infrastructure to seamlessly capture human computer-use demonstrations and transform them into state-action pairs. However, the synthetic data in these works are all relatively small in scale, and there is still a severe lack of available open-source trajectories in the CUA domain. The prohibitive cost of human verification impedes scalability, leaving the efficient synthesis of diverse and high-quality planning trajectories as a critical open challenge.

## 3. Tree-Structured Data Synthesis Pipeline

We propose **Tree-Structured Verifiable Evolution**, a multi-agent framework for synthesizing diverse trajectories to address the data scarcity constraining CUA advancement, as shown in Figure 1. To enhance trajectory diversity and quality, our framework coordinates five specialized agents across two distinct phases: tree-structured online exploration and post-hoc quality evaluation and improvement.

### 3.1. Tree-Structure Verifiable Evolution

We define the exploration manifold as a tree, where nodes encode states $s$, and edges represent transition actions $a$.

### 3.1.1. INITIALIZATION WITH WORLD KNOWLEDGE

To mitigate exploration bias and prevent premature convergence to shallow nodes, we propose World-Knowledge Initialization. By structuring official documentation into a hierarchical knowledge base $\mathcal{W}$, we classify the tasks of these applications into several categories based on shared functional dependencies. This organization provides a high-level prior that guides the agent toward semantically rich

interaction subspaces. Furthermore, merely invoking a blank application is often insufficient for deep exploration. To ensure comprehensive coverage, we formulate a hierarchical data paradigm consisting of Apps, Categories, and Trees. Our framework conducts multi-tree exploration within each Category, where each individual tree scales to 100–1,000 trajectories to ensure both breadth and depth of the interaction space. For instance, an empty IDE cannot facilitate debugging tasks. To bridge this gap, we formalize environment initialization as $s_0 = \Phi(w_c, \mathcal{P})$, where $w_c \in \mathcal{W}$ specifies a task category and $\mathcal{P}$ represents a pre-configured asset pool of files such as images, documents, accounts, and configurations. By injecting these context-specific assets, $\Phi$ transforms the environment into a non-trivial initial state $s_0$. A typical example is an IDE pre-loaded with a functional project, which establishes a necessary foundation for subsequent exploration. Furthermore, the knowledge serves as a persistent semantic context for the exploration agent at each step, ensuring that the sampled actions remain aligned with the intended functional domain and preventing the exploration from drifting into irrelevant or stochastic UI states.

### 3.1.2. ONLINE EXPLORATION

To facilitate online tree exploration, we define an exploration tuple $E_t = \langle a_t, g_t^{\text{step}}, g_t^{\text{final}}, o_{t+1}^{\text{text-exp}}, c_t^{\text{rat}} \rangle$. This tuple encapsulates the executable action $a_t$, the immediate intent $g_t^{\text{step}}$, and a dynamic hypothesis of the long-term objective $g_t^{\text{final}}$. To enhance planning coherence, each tuple also includes a textual prediction of the expected observation and an exploration rationale $c_t^{\text{rat}}$ that documents the underlying decision logic. At each node, the exploration agent $\pi_{\text{explore}}$

synthesizes $K$ candidate actions $\{E_t^{(k)}\}_{k=1}^K$ by sampling from the distribution:

$$\{E_t^{(k)}\}_{k=1}^K \leftarrow \pi_{\text{explore}}(o_t, h_t, w_c, \mathcal{M}_{\text{pre}}) \tag{1}$$

where the generation is conditioned on the current visual observation $o_t$, the world knowledge of this task category $w_c$, and global prefix histories $\mathcal{M}_{\text{pre}}$. Crucially, the policy incorporates a trajectory history $h_t$ to enforce high coherence between the agent's actions and its historical intent, thereby preventing the generated trajectories from descending into aimless or stochastic exploration. To optimize context efficiency, we retain only the textual components of the history.

**Adaptive Tree Topology**   To optimize the trade-off between functional coverage and computational efficiency, we propose a strategy that adapts the exploration tree topology to both temporal progression and semantic context. First, we implement a temporal width decay mechanism that imposes a step-dependent upper bound $K_{\max}(t)$ on the branching factor. By partitioning the trajectory into discovery, development, and convergence phases, this strategy systematically reduces exploration breadth as depth increases. This design maximizes early-stage diversity while concentrating computational resources on refining promising paths in later stages. Within this temporal framework, the exploration agent refines the specific branching factor through context-aware adjustment. When the observation $o_t$ presents multiple actionable elements, the agent prioritizes action diversity by generating candidates across distinct clusters to ensure comprehensive coverage. Conversely, if $h_t$ indicates an ongoing sub-task sequence, the agent maintains sequential consistency by restricting the exploration to essential steps, thereby preserving logical coherence. Finally, upon completing a valuable task loop, the agent synthesizes a termination action to prune the branch. This balance ensures efficient exploration, maximizing the yield of valid trajectories.

**Step Verification**   To guarantee trajectory fidelity, we employ a step-level verification agent to validate the outcome of each transition, shifting the evaluation paradigm from ambiguous outcomes to precise intermediate states. This design is motivated by the fact that while evaluating the completion of complex GUI tasks is inherently difficult without human oversight, validating atomic steps is significantly more feasible by simply checking if the interface's state change aligns with the expectation. Therefore, we employ the verification agent to assess the semantic consistency between the actual observation $o_{t+1}$ and the predicted textual observation $o_{t+1}^{\text{text-exp}}$. By classifying results into discrete states (e.g., SUCCESS, NO_CHANGE, UNEXPECTED_CHANGE), this mechanism not only filters out invalid branches to ensure data purity but also injects immediate feedback into the history, enabling the agent to perform on-the-fly error recovery.

**Global Memory**   Since trees within the same application sub-category often originate from similar initial states, a critical challenge lies in preventing redundancy across independent explorations. To address this, we propose a global memory mechanism designed to maximize functional diversity across these trees. Specifically, we maintain a global prefix memory $\mathcal{M}_{\text{pre}}$ to maximize diversity across tree explorations. This mechanism operates on the hypothesis that the core semantic intent of a CUA task is established within the initial $L$ steps. Formally, $\mathcal{M}_{\text{pre}}$ aggregates the step-wise goals from previously explored prefixes: $\mathcal{M}_{\text{pre}} = \bigcup_i \{(g_{i,0}^{\text{step}}, \ldots, g_{i,L-1}^{\text{step}})\}$. During the shallow exploration phase (where $t < L$), we impose a novelty constraint denoted as $g_t^{\text{step}} \notin \text{Semantics}(\mathcal{M}_{\text{pre}} \mid h_t)$. This explicitly directs the agent to diverge from established trajectories given identical historical contexts.

**Scalable Concurrent Exploration**   Applying tree-structured planning to general-purpose CUA environments encounters a critical bottleneck: the absence of system-level snapshotting. Real-world operating systems lack the capability for arbitrary state resets, which are essential for non-linear exploration. To address this, we develop a scalable execution engine based on deterministic node replay. Specifically, to revisit a target node $s_t$, the system resets the environment to $s_0$ and re-executes the recorded action history $A_{0:t-1}$. To handle environmental stochasticity (e.g., system clock changes), we enforce a visual consistency check between the historical and replayed observations. Furthermore, we introduce an asynchronous parallel framework, where multiple workers dynamically fetch unexplored nodes from a global queue, reconstruct states on-demand, and execute hybrid traversal strategies to maximize throughput. Detailed implementations and consistency algorithms are provided in Appendix A.

### 3.2. Quality Evaluation and Improvement

We finalize the dataset by mapping action sequences to multi-level task instructions, followed by a filtering and reasoning generation protocol that validates quality and enriches trajectories with the thinking process.

**Hierarchical Task Summarization**   We employ the summary agent to synthesize a task description $G^*$ via a bottom-up approach. By filtering out low-level redundancies, this agent abstracts the core semantic intent from the exploration trajectory. This abstraction process is stratified into two hierarchical levels. At the top level, the agent performs trajectory-level summarization to derive a global task instruction by synthesizing the semantic context from the comprehensive execution record. At the bottom level, it executes sub-trajectory extraction to partition the trajectory into coherent subtasks. Here, a sub-trajectory is defined as a

*Table 1.* Statistics of our synthetic data across different data formats. We report the counts before (**Original**) and after quality filtering (**Filtered**), along with the average action steps per instance.

| TreeCUA Data | Sample Count | | Avg. Steps | Filtering Target |
|---|---|---|---|---|
| | Original | Filtered | | |
| **Step** | 900k | 708k | 1.0 | Exec.&Replay Fail |
| **Sub-Traj** | 150k | 101k | 3.3 | Redundant Explore |
| **Long-Traj** | 100k | 50k | 18.0 | Low-quality Score |

high-value interaction segment driven by a single intent that yields a substantive environmental state change. By identifying contiguous sequences of success states and filtering out noisy steps, the agent effectively isolates these coherent stage-level units.

**Quality Evaluation and Filteration**  For data filtering, we employ a quality evaluation agent to assess the generated trajectory across the following four dimension criteria: (i) task utility, by aligning with realistic user intents to filter stochastic navigation; (ii) step efficiency, by penalizing redundant state transitions; (iii) consistency, by verifying semantic alignment between outcomes and directives; and (iv) coherence, by ensuring logical fluidity and minimizing erratic switching. For each dimension, the agent assigns a score ranging from 0 to 3 and only trajectories that surpass a threshold are retained for the final dataset.

**Reasoning Improvement**  To generate high-quality training data, we employ hindsight reasoning synthesis to reconstruct the reasoning process using the global context of completed trajectories. Unlike tentative forward reasoning, this process uses the final task target $G^*$ and the future context $f_t$, which encapsulates valid subsequent steps, to produce a defial context, progress reflection on execution histornitive reasoning chain $C_t^*$. The synthesis model operates as follows:

$$C_t^* = \Psi_{syn}(G^*, o_t, a_t, g_t^{step}, h_t, f_t)$$

The synthesized reasoning process is structured into four dimensions: observation of the visuy, planning of strategic roadmaps, and impact assessment toward the final goal.

## 4. Synthesized Data Statistics and Analysis

In this section, we present a comprehensive analysis of the synthesized dataset to validate the effectiveness of our Tree-Structured Verifiable Evolution method.

### 4.1. Dataset Overview

Leveraging the tree-structured verifiable evolution framework and our quality filtering strategy, we curate a set of 50k high-quality trajectories from an initial pool of 100k

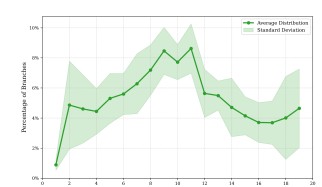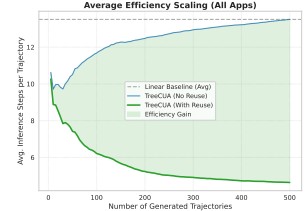

*Figure 2.* Analysis of exploration strategy. (a) The distribution of average branching factor across depths. A node with $n$ branches has a branching factor of $n - 1$. (b) Efficiency comparison benchmarking tree-structured exploration (with and without node reuse) against linear baselines.

generated trajectories. Building on this foundation, we decompose these trajectories into distinct stages based on task intentions and prune stages with redundant explorations, resulting in 101k high-quality sub-trajectories. Furthermore, by aggregating all explored nodes within the exploration tree and validating single-step execution outcomes, we compile 708k step-level training samples to construct our final dataset. Detailed statistics are presented in Table 1. We also compare tree-structured data with recent open-source CUA datasets in Table 2.

### 4.2. Tree Depth and Efficiency Analysis

A pivotal challenge in synthesizing CUA trajectories lies in arbitrating the trade-off between trajectory diversity and cost efficiency. Within our tree-structured framework, this tension is modulated by the branching depth. Branching at shallow depths maximizes diversity but incurs substantial computational overhead due to limited node reuse. Conversely, deferring branching to deeper levels enhances efficiency via extensive prefix sharing but inherently restricts diversity, risking path homogeneity. Consequently, we hypothesize that branching at intermediate depths offers the equilibrium for exploration.

First, We validate this hypothesis through a statistical analysis of the data distribution of branching depths generated by tree-structured verifiable evolution. As illustrated in Figure 2 (a), the results indicate that trajectories predominantly branch around a depth of 10, with seldom at shallow and deep nodes, thereby maintaining effective node reuse without sacrificing exploration. Also, we evaluate the efficiency of our tree-structured exploration relative to sequential baselines, measured by the average inference steps per trajectory. Figure 2 (b) shows our approach leverages node reuse to amortize exploration costs, significantly reducing the average inference steps per trajectory as the sample size grows. In contrast, sequential methods generate trajectories in isolation, leading to a linear accumulation of total cost and a constant average inference overhead.

*Table 2.* Comparison of data synthesis methods. Our method outperforms baselines by enabling fully automated, tree-structured exploration with amortized costs and dual-level verification.

| Data Source | Core Method | Topology | Exploration | Scaling Efficiency | Verification | Traj. Scale |
|---|---|---|---|---|---|---|
| OpenCUA(Wang et al., 2025) | Human Demo | Linear Chain | – | $o(N)$ | Traj.-Level | $\sim$22K |
| ScaleCUA(Liu et al., 2025) | Human + Auto | Linear Chain | Random Walk | $o(N)$ | Traj.-Level | $\sim$19K |
| **Ours** | **Fully Auto** | **Exploration Tree** | **Knowledge-Driven** | $< o(N)$ | **Stepwise + Traj.** | **50K+101K** |

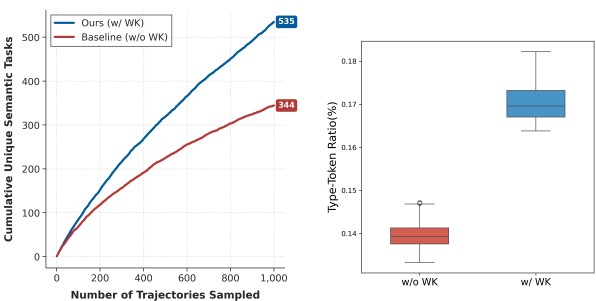

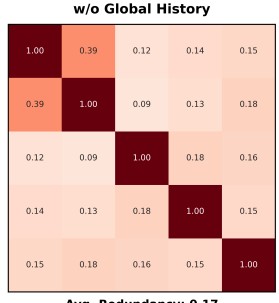

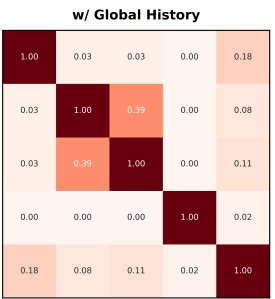

*(a)* Semantic Task Discovery    *(b)* Step-level Lexical Diversity

*Figure 3.* Impact of World Knowledge (WK) on Exploration Diversity in VS Code. (a) Semantic Task Discovery: Cumulative count of unique tasks, defined by a TF-IDF cosine similarity threshold of $< 0.65$. (b) Lexical Diversity: Type-Token Ratio (TTR) averaged over 20 random samples of 500 step-goals.

*Figure 4.* Analysis on Inter-Tree Action Redundancy. Analysis of action overlap between trees within the same setting. Redundancy is quantified via pairwise Jaccard similarity, matching actions by type and grid-quantized coordinates to mitigate pixel noise.

### 4.3. Long-tail Analysis

Beyond exploration efficiency, we investigate whether World Knowledge (WK) enables the agent to transcend the intrinsic biases of the Exploration Agent and discover "long-tail" professional functionalities. We conduct a comparative analysis in the VS Code environment, comparing an agent equipped with domain documentation (*w/ WK*) against one performing blind exploration (*w/o WK*).

We first quantify the breadth of explored tasks by calculating the cumulative number of unique semantic intents across 1,000 sampled trajectories. As shown in Figure 3 (a), we vectorize task descriptions using TF-IDF and count a task as "unique" if its cosine similarity to all previously discovered tasks is below 0.65. The results reveal that the *w/o WK* baseline quickly hits a "semantic ceiling" (344 unique tasks), as it tends to repetitively propose common operations (e.g., "opening a file" or "basic editing"). In contrast, the *w/ WK* agent maintains robust, near-linear growth in discovery (535 unique tasks). This confirms that domain knowledge acts as a semantic bridge, allowing the agent to identify and execute specialized, long-tail tasks that are rarely activated by the model's internal priors. To further examine the granularity of the agent's intent, we evaluate the linguistic diversity of step-level goals using the Type-Token Ratio (TTR). We repeatedly sample 500 step-goals 20 times to construct the boxplot in Figure 3 (b). The *w/ WK* group exhibits a significantly higher median TTR, indicating a more diverse and

precise vocabulary.

### 4.4. Diversity Analysis

To validate the efficacy of global history in minimizing inter-tree redundancy, we analyze action overlap across five sequentially generated trees per sub-category. To robustly handle pixel-level variations, we apply spatial quantization by mapping coordinates to a $20 \times 20$ grid. Actions are defined as identical if they match in type, grid location, and input text. Redundancy is then quantified using the pairwise Jaccard Similarity of unique action sets.

Figure 4 visualizes the pairwise similarity heatmaps. In the w/o Global History baseline (Left), we observe high off-diagonal similarity scores with an average redundancy of 0.17. This confirms that without historical context, the VLM's intrinsic bias drives repeated interactions with visually prominent UI elements across sessions, resulting in redundant synthesis. In contrast, the w/ Global History setting yields a significantly sparser heatmap with an average redundancy of 0.08. By recording the history of early steps, the history mechanism compels immediate divergence at the root level, ensuring that each tree explores a semantically distinct functional territory.

## 5. TreeCUA Training Recipe

In this section, we introduce the training recipe of our model, which comprises a two-stage SFT protocol for establishing

*Table 3.* Performance comparison of foundation models across different domains on the OSWorld-Verified benchmark. Results are reported as Success Rate (%).

| Model | Steps | Overall | Chrome | GIMP | Calc | Impress | Writer | Multi | OS | TB | VLC | Code |
|---|---|---|---|---|---|---|---|---|---|---|---|---|
| **Closed-source Models** | | | | | | | | | | | | |
| Seed-1.8 | 100 | 61.92 | 63.0 | 53.8 | 72.3 | 68.0 | 82.5 | 49.0 | 70.8 | 60.0 | 58.2 | 73.9 |
| Claude-Sonnet-4.5 | 50 | 58.1 | 56.4 | 57.7 | 66.0 | 57.5 | 65.2 | 47.0 | 70.8 | 66.7 | 52.9 | 69.6 |
| **Open-source Models (Online Learning)** | | | | | | | | | | | | |
| DeepMiner-Mano-7B | 100 | 41.6 | - | - | - | - | - | - | - | - | - | - |
| DART-GUI-7B | 30 | 42.1 | 52.1 | 76.9 | 19.2 | 48.8 | 60.9 | 16.7 | 62.5 | 60.0 | 39.3 | 69.6 |
| GUI-Owl-7B | - | 34.9 | - | - | - | - | - | - | - | - | - | - |
| **Open-source Models (Offline Learning)** | | | | | | | | | | | | |
| Qwen2.5-VL-7B | 50 | 5.5 | 8.7 | 11.5 | 0.0 | 0.0 | 4.3 | 1.1 | 8.3 | 6.7 | 17.6 | 21.7 |
| ScaleCUA-7B | 50 | 15.0 | - | - | - | - | - | - | - | - | - | - |
| OpenCUA-7B | 15 | 24.3 | 36.9 | 50.0 | 10.6 | 36.1 | 26.1 | 6.5 | 29.2 | 53.3 | 29.4 | 43.5 |
| UI-TARS-1.5-7B | 50 | 25.1 | 28.8 | 50.0 | 4.3 | 36.1 | 39.1 | 9.8 | 25.0 | 46.7 | 18.8 | 47.8 |
| UltraCUA-7B | 15 | 28.9 | 41.2 | 50.0 | 13.9 | 27.1 | 55.4 | 10.6 | 37.0 | 33.6 | 43.3 | 46.7 |
| DeepMiner-Mano-7B (Offline RL) | 100 | 32.7 | - | - | - | - | - | - | - | - | - | - |
| TreeCUA-7B | 50 | 34.6 | 28.3 | 76.9 | 27.7 | 40.4 | 43.5 | 14.0 | 58.3 | 33.3 | 41.2 | 47.8 |
| TreeCUA-DPO-7B | 50 | **36.6** | 39.1 | 76.9 | 25.5 | 29.8 | 47.8 | 15.1 | 54.2 | 53.3 | 47.1 | 60.9 |

foundational and cognitive capabilities, followed by our proposed TreeCUA-DPO for optimizing planning.

## 5.1. Two-Stage SFT

For two-stage SFT, we first cultivate foundational exploration capabilities, and subsequently utilize cognitive intent data to align the model with human intent.

**Stage 1: Foundational Exploration Learning.** The primary objective of this stage is to establish the model's foundational perception and planning capabilities. The training dataset comprises all filtered step-level data, alongside multi-level tasks summarized from raw trajectories.

**Stage 2: Cognitive Intent Alignment.** To better align the model with realistic user behaviors, we conduct further training using a collection of high-quality trajectories that are more consistent with realistic user tasks. Leveraging task examples written by human experts, we refine the raw tasks generated by exploration trajectories to better align with human intent. As the refined tasks might be misaligned with the raw trajectories, we employ closed-source LLMs to resample high-quality trajectories based on them for this stage.

## 5.2. TreeCUA-DPO

While SFT effectively teaches the agent how to interact with the interface, the model often struggles to distinguish between physically valid actions that serve different semantic intents. Constructing preference datasets to resolve this ambiguity is inherently complex. Due to the absence of immediate ground truth during online execution, standard

DPO (Rafailov et al., 2023) typically necessitates labor-intensive human intervention to manually identify failures and annotate corrective actions. TreeCUA-DPO turns this challenge into an asset. By leveraging our intrinsic branching topology, the framework acts as a natural counterfactual generator. It automatically contrasts successful branches against failed or suboptimal ones within the same context, yielding high-quality, physically valid preference pairs at zero additional cost.

We identify branching nodes $s_{branch}$ where the Exploration Agent explored multiple paths. For any such node with current observation $o_t$ and history $h_t$, we consider two diverging successful branches leading to distinct final goals $G_A^*$ and $G_B^*$ through actions $a_A$ and $a_B$, respectively. Tree-DPO specifically addresses this by constructing dual preference pairs:

(i) Conditioned on $G_A^*, h_t$: $(y_{win} = a_A, y_{lose} = a_B)$

(ii) Conditioned on $G_B^*, h_t$: $(y_{win} = a_B, y_{lose} = a_A)$

If both $a_A$ and $a_B$ are physically valid and successful in their respective contexts, the model cannot distinguish them by merely predicting "which action will fail." Instead, it is forced to align its policy with the specific semantic intent of the goal $G^*$. This process effectively disentangles interface affordance from user intent.

Specifically, if the step verification result $r$ for an action is negative, any preference pair employing it as the positive sample ($y_{win}$) is discarded. Furthermore, we implement a depth-uniform sampling strategy with a cap on pairs per node to ensure balanced optimization across both high-level strategic branching and fine-grained micro-interactions. We collected 16k preference pairs using the aforementioned

*Table 4.* Comparison of foundation models on our constructed OOD benchmark (Success Rate).

| Model | OOD SR (%) |
|---|---|
| Qwen2.5-VL-7B (Bai et al., 2025) | 0.7 |
| TreeCUA-7B | 26.1 |
| TreeCUA-DPO-7B | 28.1 |

method, which served as the basis for the third stage of training.

# 6. Experiments

## 6.1. Baselines

We benchmark our model against open-source models, including Qwen2.5-VL (7B) (Bai et al., 2025), UI-Tars-1.5-7B (Qin et al., 2025), as well as models trained using recent data synthesis or annotation pipelines, such as OpenCUA (Wang et al., 2025), ScaleCUA (Liu et al., 2025) and UltraCUA (Yang et al., 2025b). We adopt OSWorld-Verified (Xie et al., 2024) as our primary evaluation benchmark. We utilize official scores for models listed on the OSWorld leaderboard and reference results from original papers for those that are unlisted.

## 6.2. Effectiveness on GUI Automation

We first evaluate TreeCUA-7B and TreeCUA-DPO-7B on the OSWorld-Verified benchmark. The results in Table 3 demonstrate that our proposed TreeCUA-7B achieves superior performance compared to recent works of similar scale, such as ScaleCUA-7B (Liu et al., 2025) and OpenCUA-7B (Wang et al., 2025). Leveraging our constructed preference data, TreeCUA-DPO-7B achieves superior performance compared to TreeCUA-7B. Our results across various applications demonstrate that utilizing DPO could enhance the performance on logic-intensive and sequential tasks, such as TB, Code, and Chrome. In these domains, DPO effectively corrects reasoning errors and ensures strict adherence to steps. Crucially, as this DPO data is derived directly from previous tree-structure exploration without additional computational overhead, these results demonstrate the efficiency and effectiveness of the tree-structured verifiable evolution method.

## 6.3. Effectiveness on Generalization

To evaluate the generalization capacity of the model trained on our constructed data, we construct an Out-of-Distribution (OOD) benchmark encompassing 10 distinct applications. We synthesize test tasks by prompting LLMs with screenshots and official documentation, followed by rigorous manual verification and correction to ensure data quality. For evaluation, we employ 3 different LLMs to assess the complete interaction trajectory against the task description,

*Table 5.* Ablation study of TreeCUA training stages on the OSWorld benchmark. We evaluate the impact of different data components in a progressive manner. **SR** denotes Success Rate.

| Setting | SR (%) | Δ |
|---|---|---|
| − w/o Stage 2 | 20.8 | -13.8 |
| − w/o Stage 1 | 26.3 | -8.3 |
| **TreeCUA-7B** | **34.6** | - |

*Table 6.* Comparison with open-source data trajectories on both in-domain OSWorld evaluation and out-of-domain (OOD) test.

| Model Setting | ID OSWorld | OOD |
|---|---|---|
| Qwen2.5-VL-7B | 5.5 | 0.7 |
| + OpenCUA & ScaleCUA | 20.2 | 12.1 |
| **+ TreeCUA (Ours)** | **34.6** | **26.1** |

deeming a task successful only if it passes two consecutive evaluation rounds to ensure robustness. Crucially, to mitigate ambiguity and enhance the accuracy of this model-based evaluation, we explicitly constrain critical intermediate nodes and the required final UI state within each task description. This design ensures that task completion is directly observable via the visual interface, significantly reducing the inherent difficulty of automated evaluation. In total, we construct 306 OOD test tasks .

We first compare Tree-CUA-7B with the Qwen2.5-VL-7B as the baseline in the Table 4. The results show that our model significantly increases the OOD performance, demonstrating the effectiveness of our dataset.

## 6.4. Ablation Study

We further evaluate both ID and OOD performance across various training stage settings. As detailed in Table 5, we conduct an ablation study on the two-stage SFT protocol. Specifically, we compare models trained exclusively with exploration data (w/o Stage 2) against those trained directly on cognitive intent data (w/o Stage 1). The results reveal that bypassing either stage results in marked performance drops. This is especially detrimental when excluding cognitive intent data, as it serves to align the agent with human decision-making patterns.

Furthermore, we evaluate the effectiveness of our synthetic dataset against open-source alternatives (OpenCUA (Wang et al., 2025) and ScaleCUA (Liu et al., 2025)). To ensure a fair comparison, we conduct SFT using the same backbone model, Qwen2.5-VL-7B, across all datasets. Table 6 shows that TreeCUA consistently outperforms baselines on both ID and OOD benchmarks. These results validate that our tree-structured exploration significantly boosts both task execution performance and robust generalization.

# 7. Conclusion

In this work, we propose TreeCUA, which develops a tree-structured verifiable evolution method via multi-agent framework for efficiently scaling GUI automation. To enhance the diversity and quality, we employ multiple strategies, including world knowledge initialization, online asynchronous exploration, and diversified post-processing. Besides, by treating branching nodes as natural sources of preference data, we collect preference datasets without incurring additional costs and thus train TreeCUA-DPO. Experimental results show that our TreeCUA-7B and TreeCUA-DPO-7B significantly surpass existing models on the OSWorld benchmark and out-of-domain (OOD) tasks, demonstrating superior performance and strong generalization.

# Impact Statement

This paper presents work whose goal is to advance the field of Machine Learning. There are many potential societal consequences of our work, none which we feel must be specifically highlighted here.

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

# A. Implementation Details of Scalable Execution

In this section, we elaborate on the engineering challenges and solutions for implementing tree-structured exploration in non-reversible operating system environments.

## A.1. Deterministic Node Replay with Visual Consistency

Since standard OS environments (e.g., Windows, Linux, macOS) do not support state snapshotting like game emulators, we cannot simply reload a previous state $s_t$ to explore a new branch. We solve this via a replay mechanism.

**Replay Process.** To restore state $s_t$, the agent initiates a "hard reset" to the initial state $s_0$ (using VM snapshots or container resets) and sequentially executes the action history $h_t = (a_0, a_1, \ldots, a_{t-1})$.

**Consistency Check.** A major challenge in replay is non-deterministic visual noise (e.g., blinking cursors, changing system time, dynamic loading spinners). To ensure the replayed state $\hat{s}_t$ is semantically identical to the original state $s_t$, we employ a Root Mean Square (RMS) difference check on the screenshots:

$$\delta(o_t, \hat{o}_t) = \sqrt{\frac{1}{N} \sum_{i=1}^{N} (o_t^{(i)} - \hat{o}_t^{(i)})^2} \tag{2}$$

where $N$ is the total number of pixels. The restoration is deemed valid only if $\delta < \epsilon$. We empirically set $\epsilon = 5.0$ (for 0-255 pixel values) to tolerate minor rendering artifacts while rejecting divergent states (e.g., pop-ups, failed page loads). If the check fails, we will employ an LLM to double-check, ignoring visual changes that don't affect the actual workflow (such as rotating ads, system time, etc.), to confirm whether the replay has truly failed. The initial RMS check flags 17.1% of branches as divergent; after LLM semantic verification, the true failure rate drops to 9.49%.

## A.2. Asynchronous Parallel Framework

Building upon the replay infrastructure, our system manages tree construction through a multi-worker concurrent framework. The workflow for each asynchronous worker operates as a continuous loop, initiating with node selection. In this phase, the worker samples a candidate node $s_{t+1}$ marked as UNEXPLORED from the current tree and immediately proceeds to state restoration, invoking the deterministic replay mechanism to reconstruct the corresponding environment state $s_t$. After the restoration, the process restarts to carry out tree exploration.

To optimize efficiency, we employ a hybrid traversal strategy where the worker retains a single child node for immediate local extension, while simultaneously dispatching the remaining $K - 1$ siblings to the global system pool to accelerate parallel execution. Subsequently, the worker recursively extends the retained branch by generating candidate actions—each containing specific goals and expected visual changes—and executing a randomly selected child to validate its outcome via the Verification Agent. This local exploration loop continues until the agent explicitly signals completion, reaches the maximum depth, or is pruned due to consecutive verification failures.

# B. Quality of Reasoning Process

To validate the superiority of hindsight-generated reasoning in enhancing the model's global perspective and logical analysis capabilities, we conduct a comparative experiment using the offline AndroidControl dataset. Given the absence of mobile data in our training set, AndroidControl also serves as an OOD benchmark to evaluate the model's zero-shot generalization capabilities. In this setup, both TreeCUA and Claude-4.5-Sonnet are tasked with predicting the current step's reasoning process and executable action, conditioned on identical task goals and historical action sequences. Crucially, to strictly isolate and evaluate the quality of the reasoning process, we implement a rigorous filtering strategy that retains only the subset of samples where both models correctly predict the ground-truth action. This decouples reasoning quality from action accuracy, allowing us to evaluate the rationale without the interference of incorrect predictions.

To evaluate the quality of the generated rationales, we adopt a diagnostic framework inspired by ROSCOE (Golovneva et al., 2023), which assesses reasoning through four fundamental dimensions: Semantic Alignment, Logicality, Informativeness, and Factuality. We randomly sampled 200 reasoning chains from the filtered subset and employed Gemini-3-Flash and GPT-4o as independent expert judges to provide multidimensional scores. The results demonstrate that TreeCUA significantly surpasses Claude across all metrics, which confirms that our hindsight-augmented training effectively mitigates logical

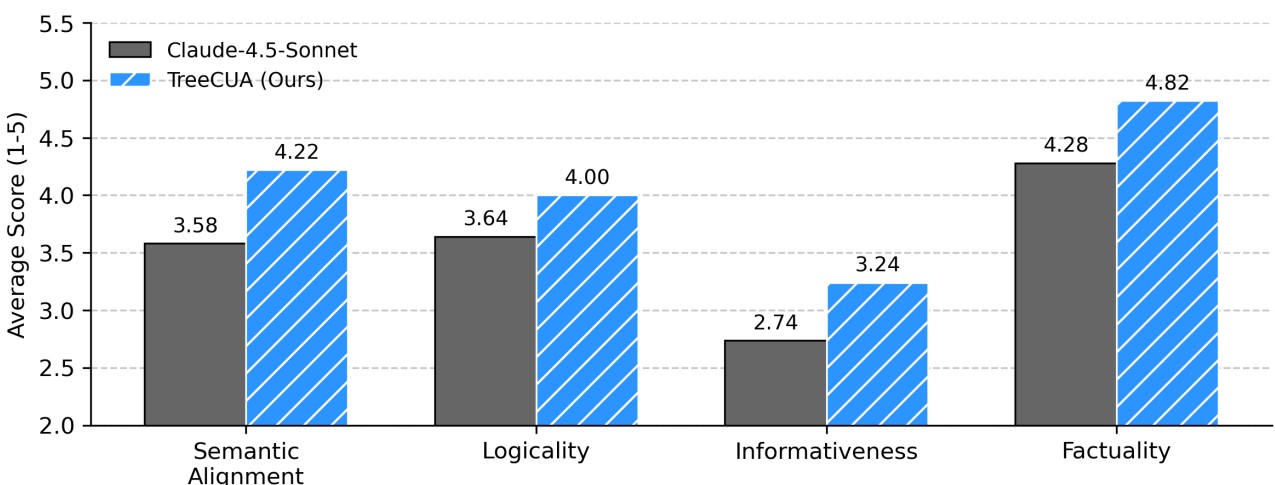

*Figure 5.* Comparison of reasoning quality between TreeCUA and Claude based on ROSCOE metrics.

leaps and tautological redundancy, resulting in a more grounded and informative reasoning process of the GUI task even in zero-shot OOD scenarios.

## C. Experimental Setup

**Data Synthesis Setup.** We utilize Claude-4.5-Sonnet (Anthropic, 2025) as the backbone for the exploration agent to ensure high-quality planning, while employing GPT-4o-mini (Hurst et al., 2024) for the auxiliary agents (verification, summarization, evaluation and reasoning) to balance cost and efficiency.

**Training Setup.** We adopt Qwen2.5-VL-7B (Bai et al., 2025) as our base model and implement a three-stage training pipeline. All experiments are conducted with a global batch size of 32.

## D. Detailed Analysis of DPO Performance Variance

To investigate the impact of the preference alignment stage, we analyze the performance shift between the SFT base model (**TreeCUA-7B**) and the DPO-finetuned model (**TreeCUA-DPO-7B**). Table 7 presents the success rate comparison across different domains, sorted by the performance gain ($\Delta$).

*Table 7.* Performance comparison between TreeCUA-7B (Stage 2) and TreeCUA-DPO-7B (Stage 3) across applications. Domains are sorted by the net performance gain ($\Delta$).

| Application | TreeCUA-7B | TreeCUA-DPO-7B | $\Delta$ | Task Characteristics |
|---|---|---|---|---|
| Impress | 40.4 | 29.8 | -10.6 | Visual Layout & Design |
| OS | 58.3 | 54.2 | -4.1 | File System Ops |
| Calc | 27.7 | 25.5 | -2.2 | Formula & Formatting |
| GIMP | 76.9 | 76.9 | 0.0 | Pixel Manipulation |
| Multi | 14.0 | 15.1 | +1.1 | Cross-App Workflow |
| Writer | 43.5 | 47.8 | +4.3 | Hybrid Editing |
| VLC | 41.2 | 47.1 | +5.9 | Menu Navigation |
| Chrome | 28.3 | 39.1 | +10.8 | Open-ended Search |
| Code | 47.8 | 60.9 | +13.1 | Env Config & Plugins |
| TB | 33.3 | 53.3 | +20.0 | Logic Rules & Filtering |

### D.1. Theoretical Attribution: Action Space and Topology

We observe a significant divergence in DPO gains, ranging from +20.0% in Thunderbird to -10.6% in Impress. This variance can be attributed to the structural alignment between the **TreeCUA exploration topology** and the **inherent interaction**

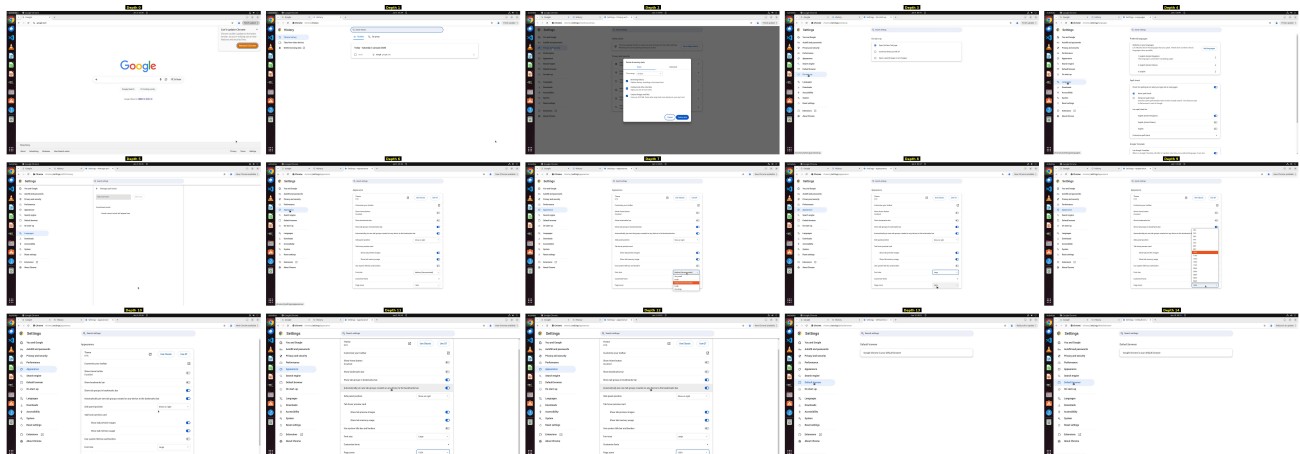

*Figure 6.* An Example of our synthesized trajectories

**logic** of the applications.

**1. Discretization of Action Space.** TreeCUA-DPO-7B achieves the most significant gains in environments dominated by **Discrete Semantic Actions** (e.g., **TB**, **Code**). In these domains, the distinction between a "correct branch" (e.g., selecting the correct menu item) and an "incorrect branch" is binary and sharp, providing strong gradient signals for preference optimization. Conversely, in domains requiring **Continuous Visual Precision** (e.g., **Impress**), where actions involve dragging elements to specific coordinates or selecting exact colors, the counterfactuals often differ only by minor continuous parameters. The DPO model may struggle to distinguish these subtle visual nuances, leading to regressions in fine-grained control.

**2. Topological Isomorpghism.** The efficacy of DPO is maximized when the application's workflow exhibits a **Hierarchical Navigation Topology**. Applications like **Chrome** and **TB** typically involve traversing through distinct states (e.g., Inbox → Email → Reply Window), which is topologically isomorphic to our exploration tree. In contrast, **In-Situ Mutation** tasks (common in **Impress** and **Calc**), where users perform multiple micro-operations within a single static view, result in shallow but wide exploration trees. This structure often fails to provide diverse high-level semantic branches for effective contrastive learning.

## E. Case study

We provide an example trajectory visualization and its corresponding first and second-step thinking, step goals, actions, and verification results in Figures 6 and 7.

---

**Example of our synthesized task**

**Task Instruction**: "Access Chrome history using 'Ctrl + H', click on 'Delete browsing data', navigate to 'On start-up' settings, go to 'Languages' settings, expand 'Customize spell check', navigate to 'Appearance' settings, change font size to 'Large', adjust page zoom to '125%', scroll down to view additional appearance settings, and navigate to 'Default browser' settings."

**tajectory**: [

{

**"step"**: 1,

**"thinking"**: {

**"observation"**: "The current interface shows the Google Chrome browser open to the Google search homepage. There is a notification in the top right corner regarding a Chrome update, but the main browser window is active and ready for input. No history or settings pages are currently open.",

**"progress"**: "This is the first step of the task. I am starting from a fresh browser state with the goal of managing browsing data and adjusting various browser settings.",

**"plan"**: "To begin the task, I will use the 'Ctrl + H' keyboard shortcut to immediately access the Chrome History page. This is the most efficient way to reach the 'Delete browsing data' option. Following this, I will proceed to clear browsing data, then navigate through the settings sidebar to configure 'On start-up', 'Languages', 'Appearance' (including font and zoom adjustments), and finally 'Default browser' settings.",

**"impact"**: "Executing 'Ctrl + H' will open a new tab displaying the Chrome History list. This will provide access to the 'Clear browsing data' link on the left-hand side, which is the next required action in the sequence."

},

**"goal"**: "Use keyboard shortcut to access Chrome history for browsing data management",

**action"**: [{"action": "key", "text": "ctrl+h"}, {"action": "wait", "duration": 2}],

**"verification_result"**: {"result_type": "SUCCESS", "feedback": "The operation successfully opened the Chrome history page as expected, showing the history list and the 'Delete browsing data' option." },

},

{

**"step"**: 2,

**"thinking"**: {....},

**"goal"**: "Click on 'Delete browsing data' to access privacy and data clearing options", **"action"**: [{"action": "left_click", "coordinate": [106, 197]}, {"action": "wait", "duration": 2}],

**"verification_result"**: {"result_type": "SUCCESS", "feedback": "The operation successfully opened the expected dialog for clearing browsing data, matching the expected outcome perfectly."}

}

]

*Figure 7.* Example of the generated task instruction and the first step trajectory of Figure 6

