# OpenReview forum: "TreeCUA: Efficiently Scaling GUI Automation with Tree-Structured Verifiable Evolution"
_ICML.cc/2026/Conference — ICML 2026 regular_

### Official Review · Reviewer_PC4J · 2026-03-09

**Soundness:** 3
**Presentation:** 3
**Significance:** 3
**Originality:** 3
**Overall Recommendation:** 5
**Confidence:** 1

**Summary:**

The paper presents a framework for computer use agents (CUA) based on tree-structured trajectories.
It provides functionalities for data synthesis and training the CUAs on the data.
A CUA can then be used for GUI automation.
Finally, the paper evaluates their approach by comparing it against existing methods.

**Compliance With Llm Reviewing Policy:**

Affirmed.

**Final Justification:**

My concerns are fully solved. I would like to keep the original rating.

**Key Questions For Authors:**

1. In table 2 you write "Linear(O(N))" and "Amortized(<O(N))" for "Cost". That confuses me a bit. Is this big-O notation? Then O(n) would mean "linear", right? And what do you mean by amortised? That wouldn't be big-O notation, right?
2. Besides comparing the runtime costs of existing approaches and your approach, wouldn't it be an honest approach to also compare for memory costs? As tree structures tend to grow large, I would expect a higher memory usage than for linear chains (sequences?).
3. You write "verifiable evolution" and explain in 3.1.2 that you do "step verification" with a verification agent. If I read verification, I usually think of theorem proving or at least model checking. How does the classification of your verification agent verify your evolution? That is not completely clear to me.

**Limitations:**

yes

**Strengths And Weaknesses:**

The paper is clearly written and well structured.
My only remark would be to please write out acronyms at least once.
DPO and SFT are never explained and it shouldn't be left to the reader to figure out what you mean by that.

I wasn't aware of the field of computer use agents before, and I am new to the topic.
As a computer scientist, I find it completely plausible to use tree structures to represent trajectories in GUIs, and I'm more surprised this hasn't been done before.
Therefore, I see the originality and soundness of the presented solutions.

---

> ### Author Rebuttal · Authors · 2026-03-31
>
> Thank you for your review. We are glad to share the same view with you that tree structures are indeed naturally well-suited for modeling GUI environments.
>
> ### Explanation of SFT and DPO
> We sincerely apologize for the oversight. Supervised Fine-Tuning (SFT) and Direct Preference Optimization (DPO) are training paradigms commonly used for Large Language Models. We will explicitly define these acronyms upon their first use and briefly explain their roles in the revised manuscript to ensure clarity for all readers.
> ### Q1. The Meaning of $O(N)$
> Yes, we are using Big-O notation. If generating one trajectory of length $L$ takes $O(L)$ inferences, generating $N$ independent linear trajectories strictly takes $O(N \times L)$. In our tree structure, multiple trajectories share the same prefix steps (which are only inferred once). Therefore, the total inference cost grows sub-linearly. We will rephrase this to "Sublinear ($< O(N \times L)$)" for mathematical precision.
>
> ### Q2. Compare for Memory Costs
> As we recover node states simply by replaying action sequences instead of saving memory-intensive virtual machine snapshots for each node, we only need to store text-based action records containing several words, making the storage cost insignificant.
> Furthermore, using a tree-based structure inherently avoids saving identical nodes repeatedly. When synthesizing the same number of trajectories, multiple branches share the same prefix nodes. As a result, the total storage cost of our tree-based approach is actually lower than that of generating independent linear chains.
>
> ### Q3. Clarification of 'Verifiable Evolution'
> We sincerely thank you for this insightful comment. We understand that in computer science, "verification" often implies formal methods such as theorem proving or model checking. To avoid potential confusion, we wish to clarify the meaning of these words in our work:
> - **"Evolution"**: refers to the process where tree-structured trajectories continuously expand and evolve through autonomous exploration within isolated sandboxes. This process is entirely independent of human expert demonstrations, enabling the autonomous scaling of data.
> - **"Verifiable"**: indicates that we utilize a Verification Agent to ensure every step of the trajectory is correct during its evolution. Specifically, the Verification Agent compares the "textual expectation of UI state changes predicted before the action" with the "actual visual screenshot obtained after the action." Only actions that pass this verification are retained in the evolution tree. This step-level validation ensures high accuracy in the autonomously expanded trajectories, thereby supporting the continuous enhancement of the agent's capabilities.
>
> We will explicitly clarify these definitions in the revised manuscript to distinguish them from formal theorem-proving verification.

---

> > ### Author Rebuttal · Reviewer_PC4J · 2026-04-01
> >
> > I thank the authors for their detailed response.
> > Maybe I'm a bit nitpicky at this point, but your answer to question one seems wrong to me, and maybe you missed my point.
> > That multiple trajectories share the same prefix still means that it is in O(N x L), as big-O notation is an upper bound.
> > It might change the average complexity, but not the worst-case complexity...
> > If you can't provide a tighter upper bound, it remains in O(N x L) and is not sublinear.
> > And I'm not aware of any notation like "Sublinear(< O(N x L))", as it simply misses the point of complexity analysis and big-O notation. If this kind of notation is common in the field of machine learning, I'm sorry to overstress this. Otherwise, I would like to encourage you to simply write the correct upper bound O(N x L) instead.
> >
> > Don't get me wrong, this is no reason to change my score, as it is just a technical detail. I will happily maintain my judgment to accept the paper.

---

> > > ### Author Response · Authors · 2026-04-02
> > >
> > > Dear Reviewer PC4J,
> > >
> > > We sincerely thank the reviewer for the follow-up. You are absolutely correct. Since Big-O notation denotes the worst-case upper bound, in extreme worst-case scenarios—where branching is minimal and trajectories are exceptionally long (i.e., a small $N$ and an extremely large $L$)—the theoretical worst-case complexity indeed remains $O(N \times L)$.
> > >
> > > Our original intent was to highlight that the empirical average cost per trajectory decreases as the number of trajectories increases due to shared prefixes. Therefore, our discussion in this section should have focused on the average-case complexity rather than the worst case, and we acknowledge that using Big-O notation to express this was imprecise. We apologize for this lack of rigor and will correct the manuscript accordingly.
> > >
> > >  We deeply appreciate your constructive and supportive attitude throughout the review process. Thank you again for your meticulous reading and for helping us improve the mathematical rigor of our paper. Wish you all the best!

---

### Official Review · Reviewer_xa8i · 2026-03-10

**Soundness:** 3
**Presentation:** 3
**Significance:** 3
**Originality:** 3
**Overall Recommendation:** 5
**Confidence:** 3

**Summary:**

This paper proposes TreeCUA, a framework for efficiently scaling training data for computer-use agents that interact with graphical user interfaces. The authors observe that GUI interaction trajectories naturally share many early steps, leading to redundant exploration when trajectories are collected independently. To address this, they organize exploration into a tree-structured trajectory space, where GUI states are nodes and actions are edges, enabling reuse of shared prefixes and more efficient branching into diverse task trajectories. The system employs a multi-agent pipeline consisting of exploration, verification, summarization, and evaluation agents to automatically generate and filter high-quality GUI trajectories. Additional mechanisms such as world knowledge guidance, global prefix memory, and adaptive branching improve exploration diversity and reduce redundant behaviors. Using the resulting tree-structured data, the authors train a GUI agent with a two-stage supervised fine-tuning protocol and further improve planning via TreeCUA-DPO, which derives preference pairs from alternative branches in the trajectory tree. Experiments on the OSWorld benchmark and an out-of-distribution evaluation show that models trained with TreeCUA data outperform existing open-source approaches, demonstrating improved task success rates and generalization.

**Compliance With Llm Reviewing Policy:**

Affirmed.

**Final Justification:**

Rebuttal has resolved my concerns and questions.

**Key Questions For Authors:**

No questions

**Limitations:**

Should discuss the bad-quality replays or trajectories percentage during data construction and analyze the cost efficiency of how many tokens are really spent on the final valid trajectories and how many are wasted during the exploration.

**Strengths And Weaknesses:**

**Strength**

1. Novel idea of using Tree representation of trajectories to efficiently produce and explore desktops, reducing redundant exploration and improving data collection efficiency compared to linear trajectory generation.
2. automatically generate, validate, and filter trajectories, reducing reliance on expensive human demonstrations and enables large-scale dataset construction for training GUI automation models.
3. uses branching trajectories as sources of preference pairs for DPO, allowing the model to learn better planning policies without requiring additional human annotations.

**Weakness**

1. Experiments lack SOTA agents comparisons, such as Qwen3-VL, UI-Venus, GUI-OWL, etc
2. Lack cost analysis, such as how many tokens spent or time spent for collection, the training GPU hours

---

> ### Author Rebuttal · Authors · 2026-03-31
>
> We sincerely thank you for your positive feedback. We are glad that you recognized the novelty of our tree-based trajectory representation, the value of our automated data generation pipeline, and the elegance of deriving DPO preference pairs directly from branching trajectories.
>
> ### W1. Missing Baselines
> Thank you for the constructive suggestion regarding SOTA baselines. As our research focuses on Data Scaling, we prioritize comparing against similar data-centric frameworks like OpenCUA and ScaleCUA to isolate the impact of our tree-structured trajectory synthesis.
>
> To ensure a rigorous and "clean" evaluation of data value, we followed the methodology of ScaleCUA by selecting Qwen2.5-VL-7B as our primary backbone. Given its relatively low native CUA performance (5.5% on OSWorld), it serves as an ideal baseline to verify the substantial gains derived specifically from our synthesized data rather than the model's pre-existing knowledge. And we mainly chose the models that were trained under the same backbone as baselines.
>
> Regarding the specific models mentioned:
> - UI-Venus: This model primarily focuses on mobile-device environments and lacks reported results on computer-use benchmarks like OSWorld, so It is not suitable for this task.
> - GUI-OWL & Qwen3-VL: We have incorporated these into our expanded comparison as the following Table.
>
> We will include these discussions and the comprehensive performance table in the revised manuscript to provide a clearer context of our work within the current landscape.
>
> | Model | Overall (%) |
> |-|-|
> | UI-TARS-1.5-7B | 25.1 |
> | Qwen2.5-VL-7B | 5.5 |
> | ScaleCUA-7B | 15.0 |
> | OpenCUA-7B | 24.3 |
> | UltraCUA-7B | 28.9 |
> | Qwen3-VL-8B | 33.9 |
> | gui-owl-7b | 32.1 |
> | TreeCUA-7B (Ours) | 34.6 |
> | TreeCUA-DPO-7B (Ours) | 36.6 |
>
> ### W2. Cost and Resource Analysis
> Thank you for pointing out the need for a detailed cost analysis. As a data-scaling framework, efficiency is a core contribution of TreeCUA. We have supplemented the manuscript with comprehensive data on API usage, token consumption, collection time, and GPU hours.
>
> 1.  Data Synthesis Cost & Token Consumption: Ensuring high-quality trajectories requires a capable exploration agent (Claude-4.5-Sonnet), which constitutes the primary expense. It actually motivated our tree-structured design: by reusing shared prefixes, we can significantly reduce API calls for exploration.
>
>     | Stage | API Calls | Input Tok | Output Tok | Cost($) |
>     |---|---|---|---|---|
>     | Exploration | 5.4 | 1.8K | 0.9K | 0.1063 |
>     | Verify | 6.2 | 6.2K | 0.1K | 0.0065 |
>     | Summary | 1 | 1.5K | 0.2K | 0.0004 |
>     | Reason | 17 | 4.1K | 0.2K | 0.0130 |
>     | Total | - | - | - | 0.1262 |
>
> 2. Cost Ablation: The Efficiency of the Tree Structure To prove the efficiency of our design, we conducted an ablation study on the synthesis cost and trajectory quality (assessed by the Eval Agent). When we remove the tree structure (reverting to standard linear exploration), the cost per trajectory triples to $0.381, and the time nearly doubles to 28 minutes. This confirms that our tree topology successfully amortizes costs by heavily reusing the expensive visual context of prefix steps. Furthermore, removing step verification drastically drops the high-quality yield rate to 41%, demonstrating that our full pipeline perfectly balances cost and data purity.
>     | Method | Time(min) | Quality(>2) | Cost($) |
>     |---|---|---|---|
>     | Full | 15 | 83% | 0.126 |
>     | w/o Tree | 28 | 81% | 0.381 |
>     | w/o Step Verify | 13 | 41% | 0.120 |
>     | w/o Replay Check | 14 | 74% | 0.124 |
> 3. Collection Time & GPU Training Hours Regarding data collection time, a single sandbox takes roughly 15 minutes to generate a trajectory. By deploying up to 100 concurrent virtual machine sandboxes, we achieved highly scalable data collection. The overall model training required roughly 1,872 H100 GPU hours across all stages
>     | Stage | GPU(80G H100 ) | Time |
>     |---|---|---|
>     | SFT-1 | 32| 41h |
>     | SFT-2 | 32 | 15h |
>     | DPO | 8 | 10h |
>
> ### Limitation 1 The Bad-Quality Replays and the Cost Efficiency Analysis
> We have provided the success rates of replay across applications and exploration depths in Figures A and B of the anonymous link (https://anonymous.4open.science/r/TreeCUA_ICML_Rebuttal-DF4E). Our findings show that the replay mechanism maintains **>90% reliability** and remains remarkably stable even as the tree deepens.
>
> Since the replay process requires no additional calls to the exploration model, it introduces negligible overhead. Only less than 10% of trajectories require one additional low-cost GPT-4o-mini call to determine replay success, as we employ a two-tier verification: RMS first flags divergent states, then an LLM performs semantic review to recover trajectories with only benign visual changes (e.g., rotating ads).
>
> For cost analysis of other stages, please refer to our response to Weakness 2.

---

> > ### Author Rebuttal · Reviewer_xa8i · 2026-04-01
> >
> > detailed additional cost analysis and performance comparison, proving the efficiency and effectiveness of the approach.

---

> > > ### Author Response · Authors · 2026-04-02
> > >
> > > Dear Reviewer xa8i,
> > >
> > > Thank you very much for your time and your highly constructive feedback. We are delighted that our detailed cost statistics and subsequent clarifications have completely resolved your initial concerns. Your deep insights have been absolutely invaluable in helping us improve the quality of this work, and we are incredibly grateful for your final support.

---

### Official Review · Reviewer_x2Jh · 2026-03-11

**Soundness:** 4
**Presentation:** 3
**Significance:** 4
**Originality:** 3
**Overall Recommendation:** 4
**Confidence:** 3

**Summary:**

This paper focuses on the data expansion problem of Computer-Use Agents (CUA). The core bottleneck currently facing the CUA field is the extreme scarcity of high-quality GUI planning trajectory data. Existing work (such as SeeClick, AMEX, OS-Atlas, JEDI, etc.) mainly focuses on the collection of GUI grounding data, i.e., identifying static UI elements in screenshots, while paying less attention to the large-scale synthesis of long-term planning trajectories (planning trajectories) that require temporal reasoning and dynamic interaction. While recent works such as OpenCUA and ScaleCUA attempt to construct planning trajectories, they still rely heavily on manual annotation, making effective expansion difficult. The authors observe that the exploration process of CUA on applications/desktops/web pages naturally exhibits a tree structure, with early functional entry points being repeatedly explored in different trajectories. Therefore, organizing trajectories into a tree structure can eliminate redundant exploration costs, while the branch nodes of the tree provide key reasoning evidence for distinguishing adjacent trajectories. This motivation is reasonable, and based on this, the authors implement a data collection, verification, and filtering pipeline executed by five automated agents, addressing the issue from the perspectives of engineering efficiency and data diversity, providing strong practical significance for CUA data. Based on this, SFT and DPO training methods were proposed, resulting in a high-performance CUA model, which proved the effectiveness of the data and training paradigm.

**Compliance With Llm Reviewing Policy:**

Affirmed.

**Final Justification:**

I have no more questions.

**Key Questions For Authors:**

* The world knowledge initialization uses the official documentation as the knowledge base, but the quality and coverage of the documentation vary greatly across different applications. Will TreeCUA's performance significantly decline for applications with poor documentation quality? Are there any related ablation experiments?

* What exactly do o_t+1 and o^exp_t+1 in online exploration data collection refer to? How are their forms defined, and what are their relationships and differences with s_t in nodes? Why does the Verification Agent validate ot instead of actions, and how is the validation implemented? Is it related to the Consistency Check in Appendix A.1?

* Regarding replay during data collection, the authors' method requires first resetting the environment to the initial state s_0 (via VM snapshot or container reset), and then sequentially re-executing the complete action history from s_0 to s_t. How to evaluate the complexity and efficiency of taking snapshots of every node versus only taking snapshots of the root node? If the load of global node snapshots is too high, are sparse snapshots feasible?

* Why are the stpes budgets different for different models in Table 3? How were different models reproduced, or did they reuse open-source data from other papers? Inconsistent budgets led to cross-sectional comparisons not being conducted under the same metrics.

* What is the Reasoning Model implemented by? The authors describe the implementations of the other four agents in Appendix C, but do not explain the source of the inference annotations. Figure 5 verifies the quality of TreeCUA inference, but the source of high-quality inference remains unknown.

**Limitations:**

* Insufficient description, analysis, and ablation of reasoning annotations in the data.

* Inadequate description of pipeline construction details; only a high-level qualitative description is provided, lacking specific details.

**Strengths And Weaknesses:**

### Strengths
* The core idea of ​​tree-structured exploration is intuitive and has engineering value. Mapping GUI exploration naturally to a tree structure effectively reduces the marginal cost of trajectory synthesis through node reuse. Figure 2(b) clearly demonstrates the efficiency advantage as scale increases.
* From world knowledge initialization, online parallel exploration, stepwise verification, hierarchical summarization to quality assessment, an end-to-end complete data synthesis pipeline is constructed, forming a complete and engineering-implementable automated process without manual annotation.
* TreeCUA-DPO utilizes the natural contrast characteristics of tree branch nodes to construct preference pair data at zero cost, and achieves decoupling of semantic intent and interface operation through bidirectional preference pairs.
* The analysis experiments are detailed. World knowledge ablation, the impact of global memory on diversity, branch depth distribution and efficiency analysis, etc., verify the effectiveness of the design from multiple perspectives. The agent model trained with the proposed data also demonstrates excellent in-domain and out-of-domain performance.

### Weaknesses
* The data synthesis cost lacks quantitative reporting. As a work whose core selling point is data efficiency, the paper fails to provide specific overhead data such as API call count, cost, and GPU time. Only Figure 2(b) shows the savings in inference steps compared to linear methods, lacking a direct comparison with OpenCUA/ScaleCUA in terms of absolute cost.
* The cost of replaying deterministic nodes (re-executing the entire path from s_0 to s_t) increases linearly with tree depth, and the robustness of visual consistency checks may degrade in complex dynamic environments. Although the paper mentions an RMS threshold ε=5.0, it lacks quantitative reporting on failure rates and recovery strategies for different applications.
* The standardization of OOD evaluation needs improvement. The self-built benchmark uses GPT-4o for evaluation and requires two consecutive passes for success, but lacks consistency verification with human evaluation. The scale of 120 test tasks (only 20 per application) is also too small.

---

> ### Author Rebuttal · Authors · 2026-03-31
>
> We sincerely thank you for your comprehensive and constructive review. Due to space constraints, we will provide further details of our data pipeline include prompts in the appendix of the final version and open-source the complete code.
>
> ### W1. Cost Statistics
> We provided detailed statistics on the API usage and time costs for data collection and the GPU time for training. And We also conducted ablation studies to evaluate each component's impact on per-trajectory time, cost, and quality (scored by Eval Agent).
>
> Due to space constraints, please refer to our **response to Weakness 2 of Reviewer xa8i** for details.
>
> Meanwhile, a direct cost comparison with OpenCUA or ScaleCUA is difficult because those methods depend on manual human effort and do not report their specific expenses.
>
> ### W2. The Failure Rates of the Replay Mechanism
> We have provided quantitative reports in Figures A and B in https://anonymous.4open.science/r/TreeCUA_ICML_Rebuttal-DF4E, detailing success rates across applications and exploration depths. Our findings show that the replay mechanism maintains **>90% reliability** and remains remarkably stable even as the tree deepens.
>
> To further ensure accuracy, we employ a two-tier verification: RMS first flags divergent states, then an LLM performs semantic review to recover trajectories with only benign visual changes (e.g., rotating ads). This effectively reduces the failure rate from **17.1%** to a final actual failure rate of **9.49%**.
>
> ### W3. OOD Benchmark
> We agree that scale is crucial for demonstrating generalization. We have expanded the OOD benchmark scale and conducted a human-consistency study. Please refer to our **response to Weakness 2 of Reviewer f6R2** for details.
>
> ### Q1. The influence of the quality and coverage of the documentation
> To empirically validate the impact of injected world knowledge, we analyzed the relationship between document quality and synthesized task performance. First, we employed three LLMs to score the documents on **Coverage** and **Quality**. Next, we sampled 1,000 generated tasks per application to evaluate Utility and Diversity.
>
> As shown in Figure C in https://anonymous.4open.science/r/TreeCUA_ICML_Rebuttal-DF4E, higher document Coverage and Quality generally yield better Task Utility and Diversity. Notably, Coverage correlates more strongly with Diversity, while Quality aligns closer with Utility. This demonstrates that broader domain knowledge directly drives task variety, whereas clear textual guidance ensures practical execution.
>
> ### Q2. The Meaning of $o^{exp}_{t+1}$
> The environmental hidden state of the OS or VM is represented as $s_t$, while $o_t$ denotes the concrete visual observation at step $t$. Before execution, the agent generates a textual prediction of the expected outcome, which we originally denoted as $o^{exp}_{t+1}$  (e.g., "A settings dialogue will appear"). The Verification Agent compares the pre-and post-execution screenshots ($o_t, o_{t+1}$) against this specific textual expectation. We prioritize verifying the actual visual outcome rather than the raw action code to ensure that each exploration step is successfully realized.
>
> To improve formal clarity, we will update this notation to o^{text\_exp}_{t+1} and refine the corresponding descriptions throughout the manuscript.
>
> This is distinct from the Consistency Check in Appendix A.1, which is strictly used to verify replay exactness
>
> ### Q3. Complexity and Efficiency of Taking Snapshots
> We appreciate this insightful suggestion. Creating snapshots for every node, which number in the millions for TreeCUA, would generate petabytes of data, incurring prohibitive storage costs and rendering high-concurrency cloud sandbox execution infeasible.
>
> In contrast, TreeCUA initializes all root nodes from a single shared image and differentiates them post-boot through lightweight configurations: uploading specific files, launching apps, or executing scripts. Using a single image incurs no storage or time overhead associated with taking snapshots, ensuring high concurrency and rapid batch deployment.
>
> We agree that the suggested sparse snapshot strategy is quite valuable, especially for ultra-long trajectories. Although it would introduce storage overhead and require re-engineering VM orchestration to maintain concurrent stability, we consider it a promising optimization for future exploration.
>
> ### Q4. Steps budgets
> As introduced in Section 6.1, we utilize official scores for models listed on the official OSWorld leaderboard and reference results from original papers for those that are unlisted. We made every effort to select the same step budgets, except in cases where they were not reported.
>
> ### Q5. Details of the Reasoning Model
> We apologize for the omission. The Reasoning Agent is implemented using GPT-4o-mini to generate the reasoning annotations offline. We will add  the exact prompt templates to the Appendix.

---

> > ### Author Rebuttal · Reviewer_x2Jh · 2026-04-02
> >
> > Thanks for the further clarification. I have no further questions. I believe this is a paper with solid design and implementation.

---

> > > ### Author Response · Authors · 2026-04-02
> > >
> > > Dear Reviewer x2Jh,
> > >
> > > Thank you very much for your time and positive feedback! We are delighted that our clarifications have fully resolved your concerns. We deeply appreciate your recognition of our framework as having a solid design and implementation, and we are grateful for your valuable input.

---

### Official Review · Reviewer_f6R2 · 2026-03-13

**Soundness:** 3
**Presentation:** 2
**Significance:** 2
**Originality:** 3
**Overall Recommendation:** 4
**Confidence:** 5

**Summary:**

TreeCUA proposes organizing GUI automation trajectories into a tree structure to reduce redundant exploration and improve training data diversity and scale. A multi-agent pipeline handles exploration, verification, summarization, and evaluation. The framework is further extended to DPO training by treating branch nodes as natural preference pairs. Experiments on OSWorld show solid gains over open-source baselines.

**Compliance With Llm Reviewing Policy:**

Affirmed.

**Final Justification:**

My concerns are fully solved. I would like to keep the original rating.

**Key Questions For Authors:**

see weakness

**Limitations:**

yes

**Strengths And Weaknesses:**

## Strengths:
- The tree topology insight is genuinely clever. Shared prefixes cut exploration cost meaningfully and the efficiency scaling analysis backs this up convincingly.
- Empirical coverage is good. In-domain OSWorld results, an OOD benchmark, and ablations all tell a consistent story.
- The world knowledge guidance analysis in Figure 3 is a nice contribution, showing the agent escapes the semantic ceiling when equipped with domain documentation.

## Weaknesses:

- The replay mechanism feels fragile. Deterministic replay under OS non-determinism such as clock changes, animations, and async loads is a real concern. The RMS consistency check feels like a bandaid and the paper never reports what fraction of branches get marked CORRUPTED in practice.

- The OOD benchmark is self-constructed with only 120 tasks across 6 apps, using GPT-4o as judge. This is thin evidence for a strong generalization claim.

- DPO hurts on some apps, notably Impress at 10.6% regression and OS at 4.1%. The theoretical explanation in Appendix D is reasonable but feels post-hoc, and no fix is proposed.

- Missed comparsion methods: DART-GUI-7B-0924, EvoCUA-8B-20260105, DeepMiner-Mano-7B, gui-owl-7b. The authors need to explain why these methods (also ~7B) are excluded from the main table. A discussion with these methods is needed.

---

> ### Author Rebuttal · Authors · 2026-03-31
>
> We sincerely thank you for the thorough and insightful review. Your detailed feedback has been very helpful in strengthening our work.
> ### W1. Replay Mechanism
> We provide a detailed statistics of replay success rates by application and tree depth (Figures A and B in https://anonymous.4open.science/r/TreeCUA_ICML_Rebuttal-DF4E).
>
> Since storing millions of snapshots would incur prohibitive storage costs and massive I/O latency, a lightweight action-recording replay mechanism is necessary for scalable, high-concurrency exploration.
>
> To mitigate the fragility of deterministic replay and handle irrelevant non-determinism, we do not rely solely on the Root Mean Square (RMS) consistency check. For cases where RMS detects a change, we then employ an LLM to distinguish between benign visual noise and actual functional state corruption (e.g., incorrect window layout or missing file content). This ensures that we only discard truly corrupted trajectories while retaining semantically valid ones. The initial RMS check flags **17.1%** of branches as divergent; after LLM semantic verification, the true failure rate drops to **9.49%**.
>
> ### W2. OOD Evaluation
> To ensure a robust generalization claim, we have **expanded our OOD benchmark to 10 applications and 306 tasks**, matching the task scale of the OSWorld. Also, we add human evaluations and **use more models** (Gemini-3-Flash, Claude-Sonnet-4.5) for alignment judging.
>
> | Method | OOD_old | OOD_new |
> | :- | :-: | :-: |
> | Qwen2.5-VL-7B | 0.8 | 0.7 |
> | + OpenCUA & ScaleCUA | 12.5 | 12.1 |
> | + TreeCUA | 26.7 | 26.1 |
> | + TreeCUA-DPO | 30.8 | 28.1 |
>
> Furthermore, we conducted a human-consistency study on 100 trajectories, where two experts independently annotated task success and discrepancies were resolved by a third expert. As shown in the following table, all judge models achieve **>95% accuracy**, confirming the reliability of our automated evaluation. This high consistency is attributable to our strict OOD task design, where completion is unambiguously determinable from the final UI screenshot alone.
>
> | Evaluator | Acc| F1 |
> |-|-|-|
> | GPT-4o | 97.00% | 96.97% |
> | Gemini 3 | 95.00% | 94.95% |
> | Claude 4.5 | 98.00% | 98.00% |
>
> ### W3.  DPO hurts on some apps
> To empirically validate our theoretical explanation in Appendix D and propose a concrete fix, we conducted a comprehensive failure mode analysis as shown in Figure D in https://anonymous.4open.science/r/TreeCUA_ICML_Rebuttal-DF4E.
>
> We employed Claude-4.5 to categorize the failure cases of both TreeCUA-7B and TreeCUA-DPO-7B into three distinct types: Localization Error, Knowledge Gap, and Planning Failure (erroneous planning or failure to recover from errors).
>
> Our analysis shows that planning failures were significantly reduced across most applications. This confirms that DPO effectively utilizes semantic branching nodes to reinforce long-horizon reasoning. In Impress and OS, the tasks depend heavily on spatial precision and specific UI knowledge. While DPO improved high-level planning, the current preference pairs do not sufficiently penalize fine-grained localization drift.
>
> To resolve this, we will expand our DPO data to cover these two types of failures.
>
> ### W4. Missed comparsion methods
> We will cite and discuss these works in the revised manuscript. We acknowledge the rapid emergence of concurrent CUA research in the field. However, as the core focus of our study is **Data Scaling**, our primary comparisons were drawn against similar data-centric methods(OpenCUA/ScaleCUA).
>
> To isolate the contributions of our synthesized data from the inherent capabilities of a foundation model, we adopted Qwen2.5-VL as our primary backbone (following OpenCUA/ScaleCUA). It is widely recognized that its inherent CUA capability is initially quite weak (scoring only 5.5 on OSWorld), which makes it an ideal, "clean" baseline to strictly isolate and verify the value of our synthesized data. Through extensive experiments on this foundation, we demonstrated significant performance gains in both ID and OOD evaluations, thereby proving the effectiveness of tree-structured data and uncovering the unique value of Tree-based DPO.
>
> In the revised paper, we will include a comprehensive comparison with these related works, as outlined in the table below. It is worth noting that DART-GUI and DeepMiner-Mano are typical online learning methods. Because they consume massive additional computational resources for real-time environmental interaction and multi-step planning during execution, they naturally yield higher performance compared to purely offline learning methods.
>
> | Model | Overall (%) |
> |-|-|
> | **Online Learning** | |
> | DeepMiner-Mano-7B | 41.6 |
> | DART-GUI-7B-0924 | 42.1 |
> | UI-TARS-1.5-7B | 25.1 |
> | **Offline Learning** | |
> | ScaleCUA-7B | 15.0 |
> | OpenCUA-7B | 24.3 |
> | UltraCUA-7B | 28.9 |
> | TreeCUA-7B (Ours) | 34.6 |
> | TreeCUA-DPO-7B (Ours) | **36.6** |

---

> > ### Author Rebuttal · Reviewer_f6R2 · 2026-04-02
> >
> > Thanks for the detailed response; my concerns are solved. Additional analysis and comparisons on algo (DPO) and clarification about the scaling scope do help.

---

> > > ### Author Response · Authors · 2026-04-02
> > >
> > > Dear Reviewer f6R2,
> > >
> > > Thank you very much for your time and continuous engagement! We are extremely glad that our additional analysis on the DPO algorithm and the clarification on the scaling scope have successfully resolved your concerns. We truly appreciate your constructive suggestions throughout this process, which have been instrumental in strengthening our paper.

---

### Decision · Program_Chairs · 2026-04-30

**Decision:**

Accept (regular)

**Comment:**

This paper proposes TreeCUA, a tree-structured framework for scalable GUI trajectory synthesis and training for computer-use agents. The core idea is intuitive and practically useful, and the paper addresses an important data-scaling bottleneck in this area.

The reviewers found the system well designed and empirically strong, with solid gains in both in-domain and OOD settings. The main concerns raised in the initial reviews were addressed satisfactorily in the rebuttal.

Overall, I view this as a solid practical/systems contribution with clear empirical value. At the same time, its main strength is engineering and data-scaling rather than a highly original methodological advance, so I recommend Accept (medium priority).